# Dissemination of local sub-variants of SARS-CoV-2 detected by detailed mutation analysis in wastewater-based epidemiology

Ryo Honda[1,2]◉*, Takashi Abe[3]◉, Tomoya Baba[4,5], Yui Araki[1], Tomohiro Kuroita[6,7], Ryo Iwamoto[6,7], Mayo Ito[8], Nami Okamura[8], Marie Kenza Yousri[8], Takashi Tamura[9], Satoshi Ezaki[9], Masanori Arita[10]

**1** Faculty of Geoscience and Civil Engineering, Kanazawa University, Kanazawa, Ishikawa, Japan, **2** Center for Infectious Disease Education and Research (CiDER), Osaka University, Suita, Osaka, Japan, **3** Department of Electrical and Information Engineering, Graduate School of Science and Technology, Niigata University, Niigata City, Niigata, Japan, **4** Joint Support-Center for Data Science Research, Research Organization of Information and Systems, Tachikawa, Tokyo, Japan, **5** Advanced Genomics Center, National Institute of Genetics, Mishima, Shizuoka, Japan, **6** AdvanSentinel Inc., Chuo-ku, Osaka, Japan, **7** Shionogi & Co., Ltd., Chuo-ku, Osaka, Japan, **8** Veolia Jenets K.K., Yokoso Rainbow Tower, Minato-ku, Tokyo, Japan, **9** Kubota Corporation, Amagasaki, Hyogo, Japan, **10** Bioinformation and DDBJ Center, National Institute of Genetics, Mishima, Shizuoka, Japan

◉ These authors contributed equally to this work.
* rhonda@se.kanazawa-u.ac.jp

## Abstract

Wastewater-based epidemiology (WBE) is effective for identifying the predominant SARS-CoV-2 variants within specific populations as well as early warning of disease outbreaks. The variant analysis in WBE has been limited to quantifying the proportion of variants, and it has been unable to trace their origins and dissemination pathways. This study aims to elucidate the emergence and transmission of locally predominant SARS-CoV-2 sub-variants through detailed mutation analysis in wastewater genomic surveillance. Genome mutations at each nucleotide position in the S region were examined to identify locally unique sub-variants in geographically distinct cities of Komatsu and Hamamatsu. Notably, the XBT variant, which had never been reported in clinical samples from Japan, was detected in wastewater in Komatsu. Moreover, a unique sub-variant of BA.5 was detected in Komatsu for a duration of 17 weeks whereas it was absent in Hamamatsu. Mutation analysis also revealed significant differences in the duration of the common BA.2.75 sub-variant's prevalence in Komatsu for 35 weeks, in contrast to only one week in Hamamatsu. These findings underscore the efficacy of wastewater-based genomic epidemiology in identifying the timing of variant entry and prevalence duration, enhancing our understanding of the origins, transmission pathways, and evolutionary trajectories of epidemically important variants.

**Data availability statement:** All relevant data are available in the paper and its Supporting Information files. The amplicon sequence data were deposited in the DNA Data Bank of Japan (DDBJ) Sequence Read Archive at https://www.ddbj.nig.ac.jp/ (accession numbers: DRR624547-DRR624567).

**Funding:** JST CREST (JPMJCR20H1), JSPS KAKENHI Grants (21KK0073), JST Program Manager (PM) Development and Promotion Program, Grants by Hiramoto-Gumi Inc. and I-Tec-Muramoto Co. Ltd. We declare that all the funders had no role in study design, data collection and analysis, decision to publish, or preparation of the manuscript.

**Competing interests:** The authors have declared that no competing interests exist.

## Introduction

Wastewater-based epidemiology (WBE) is recognized as an efficient population-scale surveillance of disease outbreaks [1]. WBE is globally used to monitor national and regional COVID-19 trends. WBE is not limited to tracking epidemic trends by quantifying gene markers of target pathogens but also identifies the predominant genotypes or variants via genomic sequencing analyses. Several studies have successfully detected and identified SARS-CoV-2 variants in wastewater [2–7]. An important advantage of wastewater genomic surveillance is its lower sampling bias, which provides a more accurate reflection of the actual proportion of variants in a population. Clinical genomic surveillance typically focuses on a limited number of selected clinical samples, potentially overlooking variants present in the broader population [3]. In contrast, since wastewater genomic surveillance encompasses the entire population served by a wastewater treatment plant, it is expected to detect all prevalent variants in a target population. However, the variant sequences obtained via wastewater genomic surveillance are heterogeneous and unsuitable for assembly into a complete genome sequence, which is essential for accurate variant lineage identification [2–3]. Conversely, clinical genomic surveillance facilitates the complete sequencing of clinical samples, which typically contain a single variant, enabling lineage identification and tracing of the origin and spread of variants of concern through haplotype analysis [8–11]. In contrast, SARS-CoV-2 genome sequences obtained from wastewater usually comprise only partial genome regions (e.g., the S region) or an array of multiple partial regions across the entire genome, making it difficult to assemble the entire genome and identify each variant present in a wastewater sample [2–5]. Consequently, wastewater-based genomic epidemiology has so far been limited to quantifying the proportion of variant lineages based on the signature combination of amino acid variations from the obtained partial genome sequences. However, solely quantifying variant proportions is insufficient to trace the origin and dissemination pathway of locally predominant variants. Although viral genome sequences in wastewater originate from multiple variants, the proportion of amplicon reads is assumed to reflect the relative abundance of these variants. Hence, by detailed comparison of single-nucleotide variants (SNVs) with haplotype data of clinical genome surveillance, the origin and behaviors of the locally predominant SARS-CoV-2 variants could be traced more precisely.

This study aimed to clarify the origin and behaviors of locally predominant SARS-CoV-2 sub-variants using wastewater genomic surveillance data. Detailed mutation analysis was applied to amplicon sequence data obtained from wastewater samples from two cities, in order to clarify the differences of locally predominant sub-variants and their time-series behaviors. The findings of this study demonstrate the utility of wastewater-based genomic epidemiology in tracing the dissemination pathway of locally predominant SARS-CoV-2 variants.

## Materials and methods

### Wastewater sampling

Influent wastewater was collected from two wastewater treatment plants (WWTPs) located in Komatsu City and Hamamatsu City, Japan. The target sewersheds served

40% of Komatsu's population (106,000) and 57% of Hamamatsu's (791,000). Total 38 wastewater samples were collected by grab sampling during the morning peak flow in Komatsu and in the 33rd week in Hamamatsu, and by 24-hr equal-volume composite sampling at 30-min intervals in the other weeks in Hamamatsu (S1 Table). Sampling was conducted twice monthly from February to November 2023. Samples in Hamamatsu were stored at -20°C after sampling pending subsequent analysis, while samples in Komatsu were subjected to the subsequent analysis within 24 hours of sampling.

## Sample processing and quantification of viral RNA

Prior to RNA extraction, 40 mL of each sample was concentrated via polyethylene glycol (PEG) precipitation without separating suspended solids [12]. Samples in Komatsu were stored at -20°C after the PEG precipitation until the subsequent viral RNA extraction. Viral RNA was extracted from the PEG-precipitated samples using the COPMAN DNA/RNA extraction kit for wastewater (AdvanSentinel, Tokyo) [13]. The RNA extracts were subjected to reverse-transcription (RT) using a random primer set and PrimeScript RT Master Mix (Perfect Real Time) (Takara Bio, Kusatsu). SARS-CoV-2 titers in the obtained cDNA were then quantified by real-time PCR with CDCN1 primer-probe using Probe qPCR Mix with UNG (Takara Bio, Kusatsu). The detection efficiency of pepper mottle mosaic virus (PMMoV) was confirmed as process control according to Alamin et al., 2022 [12].

## Targeted amplicon sequencing

Partial sequences of the spike-protein (S) region in the SARS-CoV-2 genome were obtained through targeted amplicon sequencing [5] (S1 Fig). Briefly, 13.5 μL of the RNA extract was subjected to a 20-μL reverse transcription reaction using the Reliance Select cDNA Synthesis Kit (Bio-Rad, CA, USA) and the SARS-CoV-2 spike-protein gene primer S008 (5'-AGTTGAAATTGACACATTTG-3′), according to the manufacturer's protocols. The thermal conditions were as follows: 50 °C for 60 min, 95 °C for 1 min, and 4 °C until the next step. Then, 20 μL of the cDNA samples were subjected to nested PCR reaction using KOD One PCR Master Mix (TOYOBO, Osaka, Japan). In the first PCR, the partial region of the receptor binding domain (RBD) (329–535 aa) in the S region of the SARS-CoV-2 genome was amplified using KOD One PCR Master Mix and primers S012 (5'-CAACCAACAGAATCTATTGTTAG-3′) and S008. Reaction solutions were prepared according to the manufacturer's guidelines, and the reaction volume was 41.2 μL, including 20-μL cDNA. The PCR thermal cycling conditions were as follows: 98 °C for 3 s, followed by 10 cycles of denaturation at 98 °C for 10 s, annealing at 55 °C for 5 s, and extension at 68 °C for 2 s. In the second PCR step, using 3.4 μL of the reaction mixture of the first PCR as the template, the gene sequences of the internal region (337–504 aa) were amplified using KOD One PCR Master Mix and the primers S013 (5'-TCGTCGGCAGCGTCAGATGTGTATAAGAGACAGTTCCTAATATTACAAACTTGTGC-3′) and S009 (5'-GTCTCGTGGGCTCGGAGATGTGTATAAGAGACAGACTACYACTCTRTATGGTTGGT-3′). The PCR thermal cycling conditions were as follows: 98 °C for 3 s, followed by 45 cycles at 98 °C for 10 s, 61 °C for 5 s, and 68 °C for 2 s. Amplicons were purified using Agencourt AMPure XP beads (Beckman Coulter, Brea, CA, USA) and eluted with 10 mM Tris-HCl (pH 8.5). The S gene amplicon was index-tagged using Nextera XT Index Kit v2 (Illumina, CA, USA) and purified using Agencourt AMPure XP beads according to the manufacturer's instructions. The size and DNA concentration of the purified pooled DNA libraries were measured using 4200 TapeStation (Agilent Technologies, CA, USA), and the libraries were sequenced using the Illumina MiSeq system with 300-base paired-end reads. The target sequence depth was 20,000–30,000 read pairs per sample. Among the total 38 samples collected, amplicon sequence data were successfully acquired from 21 samples (S1 Table). The amplicon sequence data were deposited in the DNA Data Bank of Japan (DDBJ) Sequence Read Archive under accession nos. DRR624547–DRR624567.

## Sequence data analysis

The amplicon sequences were processed using fastp v0.32.2 for quality filtering and paired-end sequence merging [14]. In this study, the amplicon sequence lengths exceeding or equal to 500 bp were included. Sub-lineage assignment for

the sequences was performed using Nextclade v3.4.0 [15] with manual correction. To compare with known sequences sampled in Japan, total 92,128 sequences of collection year 2023 were downloaded from GISAID [16] on 13 August 2024. The RBD regions were detected by aligning the known sequences with the reference sequence (GenBank MN908947.3) using mafft v7.526 [17]. Next, clustering was performed using cd-hit v4.8.1 [18] with 100% sequence identity and a coverage threshold of 90%. Considering the maximum error rate (2%) for sequencing [5], clusters up to 550 bp in length were extracted as sub-variants and sequentially numbered with a prefix C. These identifiers were unique to this study. A phylogenetic tree of the sub-variants was constructed using the Neighbor-Joining method [19]. The percentage of replicate trees in which the associated taxa clustered together during the bootstrap test (500 replicates) is shown next to the branches [20]. The tree is drawn to scale, with branch lengths proportional to the evolutionary distances used to infer the phylogenetic tree. Evolutionary distances were calculated using the Maximum Composite Likelihood method [21] and are expressed as the number of base substitutions per site. All ambiguous positions were removed for each sequence pair using pairwise deletion option. The final dataset consisted of 504 aligned positions between the primers S013 and S009 [5] (S1 Figure), and included 33 sub-variants sequences along with reference sequences (GenBank MN908947.3 and OY146289.1) (S1 Appendix). Evolutionary analyses were conducted using MEGA11 [22]

## Results and discussion

### Overall trend of SARS-CoV-2 titers and variant proportions.

SARS-CoV-2 titers in wastewater during the target period ranged from 580 to 30,000 copies/L in Komatsu and from 1,200–23,000 copies/L in Hamamatsu (Figs 1A and 1B). According to the GISAID data, clinical samples from Japan exhibited a shift in dominant SARS-CoV-2 variants from BA.5 and BA.2.75 (weeks 1–10) to a combination of omicron strains, including XBB.1.5, XBB.1.16, and XBB.1.91 (weeks 11–35), ultimately converging into EG.5 and BA.2.86 (S2 Fig). Dominant variants detected in wastewater mostly exhibited similar transitions as observed in clinical samples (Figs 1C and 1D). Notably, the XBT variant, previously reported in Europe (S3 Fig) but never in Japan, was detected in wastewater in Komatsu City during the 14th, 16th, and 23rd weeks. The XBT detected in this study had distinct mutations of R346T, L452R, and F486V from other XBB variants (S3 Table), which reportedly enhanced infectivity and immune evasion capability [23–25]. The XBT variants were identified in 3.1% (389/12572) and 4.6% (595/12980) reads in the 14th and 16th weeks in Komatsu, respectively (S2 Table). The detected prevalence of XBT significantly surpassed the 2% estimated error range of the amplification and sequencing method [5], highlighting its presence in the population, which clinical genomic surveillance had overlooked. Moreover, even within the lineage commonly detected in wastewater of both cities, phylogenetically distinct sub-variants were locally predominant in each city, as described in the next section.

### Detection of local sub-variants and their epidemics from wastewater

Nucleotide-level mutation analysis of the 21 amplicon sequences from wastewater identified the presence of locally predominant sub-variants and their epidemic durations in each city. For instance, several common omicron sub-variants (i.e., XBB.1.5 sub-variant C63, XBB.1.16 sub-variant C3, and EG.5 sub-variant C402) were prevalent in both cities during the same period. These sub-variants were phylogenetically close to the SARS-CoV-2 variants reported predominantly in clinical samples in Japan. Hence, detection of these omicron sub-variants in wastewater reflected their prevalence in the target cities. Moreover, the detailed mutation analysis revealed differences in epidemic durations of some common sub-variants between the two cities. The BA.2.75 sub-variant C25 was detected both in Komatsu and Hamamatsu; however, its epidemic durations varied between the two cities. In Komatsu, the BA.2.75 sub-variant C25 detected for 35 weeks (from 5th to 40th week), while it was detected in Hamamatsu only in the 9th week (S2 Table). Such difference of epidemic duration by variants and regions were also reported in clinical data of Japan [9]. According to the clinical variant data in GISAID, BA.2.75 was reported until 25th week in in Ishikawa Pref, where Komatsu City is located, while it was

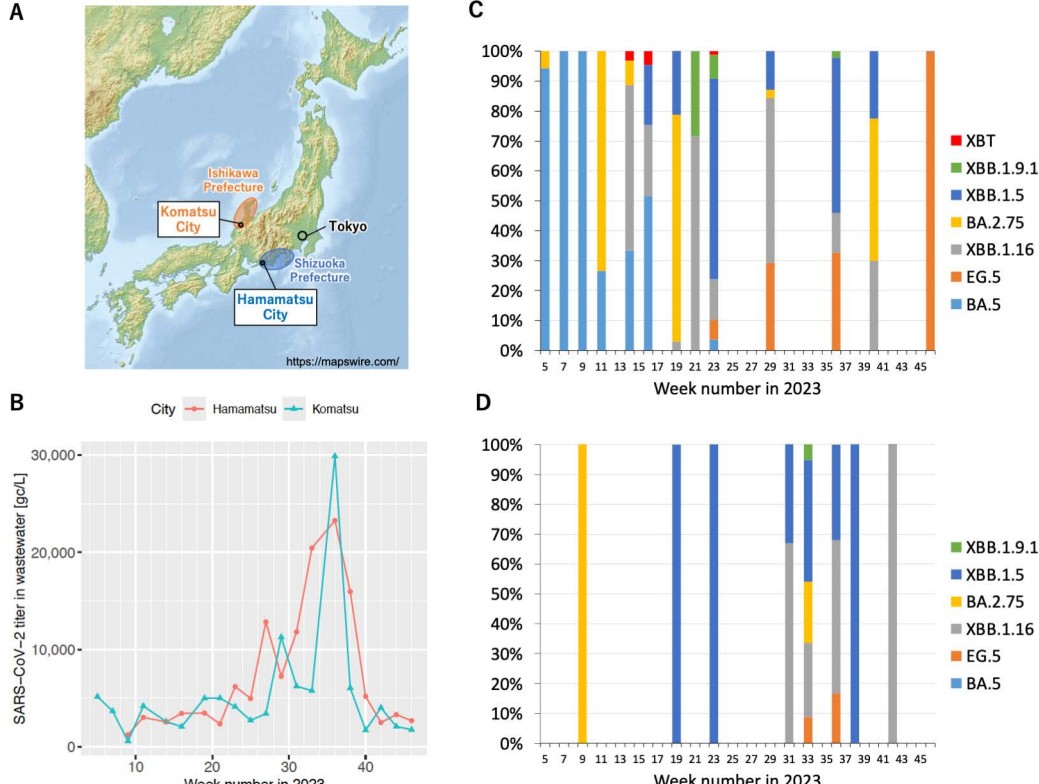

**Fig 1. Time-series change of SARS-CoV-2 titers and variant compositions in wastewater of the target cities.** (A) Locations of the target cities. (B) SARS-CoV-2 RNA concentrations in wastewater of Komatsu and Hamamatsu in 2023, as gene copies (gc) per liter of wastewater. (C) Proportion of SARS-CoV-2 variants by PANGO lineage in Komatsu. (D) Proportion of SARS-CoV-2 variants by PANGO lineage in Hamamatsu. The blank weeks in the proportion of SARS-CoV-2 variants indicate that no sequence data were obtained for the corresponding week. The map was edited from the base map data provided by Mapswire.com.

continuously reported until 12th week in Shizuoka Pref., where Hamamatsu City is located (S4 Table). In Japan, regional differences of epidemic variants are probably caused by human mobility and public health policies. Particularly, migration of a variant with high infectivity from another city would facilitate replacement of predominant variants [26]. In this study, the sub-variants detected in wastewater were phylogenetically close to the clinically predominant variants in the prefecture where the target city is located, and their epidemic durations were similar to those of the clinically predominant variants (S2 Table). These results suggest that applying detailed mutation analysis to WBE enabled the identification of regional differences in epidemic durations by variant at the city level, with a geographically higher resolution than at the prefecture level. Interestingly, locally unique sub-variants were also identified from the phylogenetic analysis. For example, BA.5 sub-variant C18 was detected only in Komatsu from the 7th week to the 23rd week, but not in Hamamatsu (S2 Table). This BA.5 sub-variant C18 was also detected widely in clinical samples from Japan (Fig 2). Hence, this BA.5 sub-variant C18 was possibly introduced from another Japanese city to Komatsu and became predominant but did not spread to Hamamatsu. These findings illustrate the efficacy of detailed mutation analysis in continuous wastewater surveillance to track variant entry, prevalence, and dissemination pathways at the sub-lineage level.

Importantly, detailed mutation analysis of the amplicon sequences suggested that epidemic sub-variants in each city are evolutionarily distinct. For example, EG.5 sub-variant C65601, XBB.1.16 sub-variant C65598, BA.2.75 sub-variants C661 and C676 were detected only in Komatsu (Fig 2). Although the absence of these sub-variants in the other city might

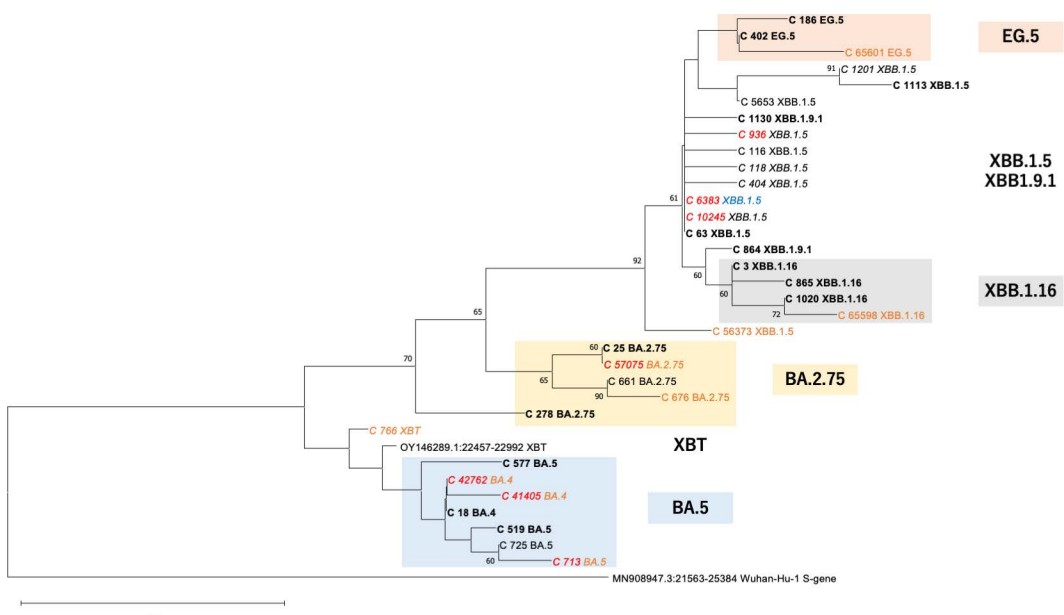

**Fig 2. Evolutionary relationships of SARS-CoV-2 sub-variants detected in wastewater.** Sub-variants matching the genomic surveillance data (GISAID) of SARS-CoV-2 viruses isolated from clinical specimens in Japan are shown in **bold**, while sub-variants with no matching data are shown in *italic*. Sub-variants detected only in Hamamatsu City are shown in **blue**, and those detected only in Komatsu City are shown in **orange**. Sub-variants with stop codons or frameshifts in the sequence are shown in **red**. Branches corresponding to partitions reproduced in less than 50% of bootstrap replicates are collapsed. Bootstrap values are indicated at the branches on a 0.01 scale.

be caused by sampling bias and sequencing failure, emergence of the local sub-variants was also supported by comparison with clinical variant data reported in the GISAID database. Most of the detected local sub-variants were phylogenetically close to the predominant clinical variants in Japan. According to the phylogenetic analysis (Fig 2), the EG.5 sub-variant C65601 was diverged from C402, which were the most predominant EG.5 variant in Japan. Similarly, BA.2.75 sub-variants C676, C661, and C57075 are indicated to evolved from the same origin of C25, which were the most predominant BA.2.75 variant in Japan, respectively. These results demonstrate that predominant variants in each target sewershed consisted of multiple sub-variants rather than a single sub-variant, suggesting emergence of unique sub-variants driven by local evolutionary processes.

Notably, some sub-variants that were not reported in clinical specimens were detected in wastewater, e.g., several XBB.1.5 sub-variants and BA.5 sub-variants C42762, C41405, and C713. Locality of sub-variants within the same lineage was also evident in XBB.1.5, where C56373 was uniquely detected in Komatsu whereas C6383 was detected only in Hamamatsu. Similarity in nucleotide sequences of these two sub-variants was relatively low (Fig 2), suggesting occurrence of locally unique mutations and dissemination of the local sub-variant. Interestingly, the local mutation included stop codons and frameshifts (S3 Table) at a higher ratio than the reported sequencing error [5]. Because sub-variants with such stop codons and frameshifts were not consistently prevalent in clinical samples, detection of these mutations in wastewater would provide crucial clues to understand genomic evolution of SARS-CoV-2. For example, in Komatsu, the appearance of BA.5 sub-variants C41405 and C42762, both with a frameshift mutation, in the 9th week implied a transition of the dominant variant from BA.5 to BA.2.75 (S2 Table, S3 Table, Fig 1C). Likewise, implicit transitions to XBB.1.5 could be observed after the appearance of the BA.5 sub-variant C713 with a stop codon in the 14th week, to XBB.1.9.1 after the appearance of the BA.2.75 sub-variant C57075 with a frameshift mutation in the 19th week, and to EG.5 with the appearance of the XBB.1.5 sub-variant C936 with a stop codon in the 23rd week, respectively. In Hamamatsu, the XBB1.5

sub-variants C6383 and C10245 with frameshift mutations appeared in the 19th and 23rd week, respectively, followed by an apparent transition to XBB.1.16 (S2 Table, S3 Table, Fig 1D). In genomic studies of clinical samples, Omicron lineages have been reported to exhibit a higher frequency of nonsense mutations, including frameshifts and stop codons, compared to the Alpha, Beta, and Delta lineages, suggesting that these mutations may facilitate the adaptation of the variant [27–29]. While the role of nonsense mutations in viral adaptation remains unclear, their higher prevalence in Omicron suggests they may be more tolerated or result from the variant's rapid evolution [28]. Detection of nonsense mutations, such as frameshifts and stop codons, in SARS-CoV-2 genomic sequences in wastewater has been reported in a previous study [30]. However, such nonsense mutations are rarely identified by clinical genomic surveillance because clinical genomic surveillance primarily relies on samples from symptomatic and diagnosed patients only. In this study, we demonstrate for the first time the potential for elucidating the mechanisms driving the transition of epidemic lineages due to the emergence of frameshifts and stop codons through nucleotide-level monitoring of Omicron sub-lineages. Furthermore, applying haplotype analysis to wastewater-based epidemiology (WBE) enables effective detection of these mutations, providing insights into more detailed evolutionary trajectories of the virus.

## Implications and limitations

In summary, wastewater genomic surveillance enables not only the identification of variant trend overlooked by clinical genomic surveillance but also the detection of sub-variants with locally unique genomic shifts through the detailed mutation analysis of amplicon sequences. Compared to clinical genomic surveillance, wastewater genomic surveillance offers several advantages: (i) more efficient and frequent surveillance (e.g., on a weekly basis) due to the smaller number of samples required; (ii) better representativeness of the actual proportions of variants in a target area; and (iii) a lower likelihood of overlooking epidemic variants. These benefits complement clinical genomic surveillance, which requires a large number of samples and is subject to significant biases in sample selection. Recent studies reported that wastewater genomic surveillance could detect emerging variants earlier than through clinical surveillance [5,31,32]. Moreover, deep sequencing of the RBD region detected cryptic SARS-CoV-2 lineages that were not found in any global databases [33]. The findings of this study also demonstrate that the utility of wastewater genomic surveillance for detection of SARS-CoV-2 sub-variants that have been uncovered by clinical surveillance. Furthermore, detailed mutation analysis in wastewater genomic surveillance facilitates the early detection of emerging local variants and the identification of their origins and dissemination pathways by comparing them with those in other countries and regions. In this study, haplotype analysis was applied to a short region within the RBD of the S gene. However, recent advancements in sequencing techniques have enabled the analysis of whole genome sequences of SARS-CoV-2 from wastewater [34–36]. Applying haplotype analysis to whole-genome sequences obtained from wastewater is expected to provide a more comprehensive view of mutation trajectories, enhancing our understanding of viral evolution, including the emergence of nonsense mutations and asymptomatic variants.

Since genetic mutations occur randomly, not all mutations are beneficial for viral replication; they also include deleterious and neutral mutations, the latter of which are neither advantageous nor disadvantageous. Strains carrying deleterious mutations are eliminated through selection, whereas those with neutral mutations persist and contribute to the formation of diverse sub-variants in different regions. In contrast, variants identified in clinical settings primarily possess mutations that enhance human infectivity (i.e., beneficial for replication). Consequently, both clinical and wastewater genomic surveillance that rely on existing catalogs of known clinical variants face challenges in capturing the evolutionary pathways leading to the emergence of novel viral lineages. To address this limitation, this study employed a nucleotide-level mutation analysis, including synonymous substitutions, rather than focusing solely on amino acid changes. This approach enables a more comprehensive understanding of viral variant diversity, particularly neutral mutations, and represents a novel and advantageous method compared to conventional approaches. By incorporating such analyses, we can better elucidate the evolutionary dynamics of the virus and the detailed trajectories leading to the emergence of new clinical variants.

Despite these advantages, several limitations of wastewater genomic surveillance should be noted. The representativeness of wastewater genomic surveillance primarily depends on the sampling methodology. Grab sampling, often conducted during morning peak flow, may fail to detect variants discharged at other times. Composite sampling, which collects multiple samples over a 24-hour period, improves representativeness but may still miss variants released between sampling intervals. Additionally, improving sample storage and processing methods has been suggested to enhance the reliability of mutation analysis in wastewater genomic surveillance. In this study, sequence data was successfully obtained in only 21 of 38 samples (55%) (S1 Table). The primary cause of sequencing failure was unsuccessful amplification in the second PCR of the nested PCR process. Although degradation of viral RNA during the transit time of wastewater in sewer lines are limited [37], RNA degradation might occur during the freeze-thaw process of the samples. The wastewater samples from Hamamatsu City were stored at -20°C and processed by PEG precipitation after thawing, while those from Komatsu City were concentrated by PEG precipitation before freezing at -20°C. This likely contributed to the lower sequencing success rate in samples from Hamamatsu samples (8/18, 44%) compared to Komatsu samples (13/20, 65%). Moreover, RNA degradation due to sample storage was found to be dependent on the initial viral RNA concentration rather than duration of storage and followed the first-order reaction kinetic (S4 Figure). These findings suggest that viral RNA degradation might be facilitated by the thawing process, which caused bacterial cell destruction and release of RNase from the bacterial cells. In addition to the possible degradation of viral RNA, presence of PCR inhibiting substances can also lead to amplification failure, particularly when suspended solids were not excluded from the sample [12]. Therefore, improving sample storage conditions and excluding suspended solids in the virus concentration step would enhance the reliability of detailed mutation analysis in wastewater genomic surveillance.

## Conclusions

Nucleotide-level mutation analysis was performed on SARS-CoV-2 genome sequences obtained from wastewater in two geographically distinct cities. Genotyping of wastewater sequence data revealed the presence of the XBT variant, which had never been reported in clinical samples from Japan. Haplotype analysis further identified locally unique sub-variants in each city, suggesting that the epidemic sub-variants in these locations are evolutionarily distinct. Notably, several sub-variants detected in wastewater carried nonsense mutations, such as frameshifts and stop codons, which were rarely identified through clinical genomic surveillance. Consequently, applying haplotype analysis to WBE is expected to elucidate detailed evolutionary trajectories, including the emergence of nonsense mutations and asymptomatic variants, and to enhance our understanding of viral evolution.

## Supporting information

**S1 Appendix.  FASTA-format sequences of 33 sub-variants identified in this study, along with reference sequences.**
(TXT)

**S1 Fig.  Mutations of SARS-CoV-2 variants in the spike-protein (S) gene.** (A) The target region of the S gene. (B) Phylogenetic tree of the major variants. (C) mutations of the major variants in the target region.
(TIF)

**S2 Fig.  Proportion of SARS-CoV-2 variants in clinical samples of Japan reported in GISAID.**
(TIF)

**S3 Fig.  Proportions of XBT variants in clinical samples registered in GISAID.** (A) Proportion in the world. (B) Proportion by country.
(TIF)

**S4 Fig. Reduction of viral RNA concentration before and after sample storage.** (A) Log reduction of SARS-CoV-2 by the initial concentration. (B) Log reduction of PMMoV by the initial concentration. (C) Log reduction of SARS-CoV-2 by duration of storage. (D) Log reduction of PMMoV by duration of storage. The SARS-CoV-2 RNA concentrations were quantified with CDCN1 assay.
(TIF)

**S1 Table. Information of samples and sequence data.** (a) Hamamatsu City. (b) Komatsu City.
(XLSX)

**S2 Table. Sequence reads of each sub-variants detected in wastewater of the target cities and the number of registrations from the clinical specimen in the GISAID database.**
(XLSX)

**S3 Table. Mutations of SARS-CoV-2 sub-variants detected in wastewater of the target cities.**
(XLSX)

**S4 Table. Number of clinical sub-variants reported in GISAID in the whole Japan, Ishikawa Prefecture, and Shizuoka Prefecture.**
(XLSX)

## Acknowledgments

The authors are thankful to Hamamatsu City and Komatsu City for their cooperation in provision of wastewater samples. Computations were partially performed on the NIG supercomputer at ROIS National Institute of Genetics.

## Author contributions

**Conceptualization:** Ryo Honda.

**Data curation:** Takashi Abe, Tomoya Baba.

**Formal analysis:** Takashi Abe, Tomoya Baba.

**Investigation:** Yui Araki, Tomohiro Kuroita, Ryo Iwamoto, Takashi Tamura, Satoshi Ezaki.

**Methodology:** Ryo Honda, Takashi Abe, Tomoya Baba, Tomohiro Kuroita, Ryo Iwamoto.

**Project administration:** Ryo Honda, Masanori Arita.

**Resources:** Mayo Ito, Nami Okamura, Marie Kenza Yousri, Takashi Tamura, Satoshi Ezaki.

**Supervision:** Masanori Arita.

**Visualization:** Takashi Abe, Tomoya Baba.

**Writing – original draft:** Ryo Honda, Takashi Abe, Tomoya Baba.

**Writing – review & editing:** Tomohiro Kuroita, Ryo Iwamoto, Masanori Arita.

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
