## [Decision Letter · Decision Letter 0]

3 Feb 2025

PONE-D-24-59114Dissemination of Local Sub-Variants of SARS-CoV-2 Detected by Detailed Mutation Analysis in Wastewater-Based EpidemiologyPLOS ONE

Dear Dr. Honda,

Thank you for submitting your manuscript to PLOS ONE. After careful consideration, we feel that it has merit but does not fully meet PLOS ONE’s publication criteria as it currently stands. Therefore, we invite you to submit a revised version of the manuscript that addresses the points raised during the review process.

The reviewers has raised sevral issues related analysis and interpretation of the results. Therefore I recomonded carefully revised on the analysis and discussion section so the findings can be more understanding for wider audience. 

We look forward to receiving your revised manuscript.

Kind regards,

Dhammika Leshan Wannigama, MD PhD

Academic Editor

PLOS ONE

Journal Requirements:

2. Thank you for stating the following financial disclosure: [JST CREST (JPMJCR20H1), JSPS KAKENHI Grants (21KK0073), Grants by civil engineering companies of Hiramoto-Gumi Inc. and I-Tec-Muramoto Co. Ltd].

3. Thank you for stating the following in the Acknowledgments Section of your manuscript: "This study was supported by the JST CREST grant (JPMJCR20H1), the JSPS KAKENHI grant (21KK0073), and grants by Hiramoto-Gumi Inc. and I-Tec-Muramoto Co. Ltd. The authors are thankful to Hamamatsu City and Komatsu City for their cooperation in provision of wastewater samples."

Please remove any funding-related text from the manuscript and let us know how you would like to update your Funding Statement. Currently, your Funding Statement reads as follows:[JST CREST (JPMJCR20H1), JSPS KAKENHI Grants (21KK0073), Grants by civil engineering companies of Hiramoto-Gumi Inc. and I-Tec-Muramoto Co. Ltd].

4. In the online submission form, you indicated that your data is available only on request from a third party. Please note that your Data Availability Statement is currently missing [the name of the third party contact or institution / contact details for the third party, such as an email address or a link to where data requests can be made]. Please update your statement with the missing information.

Reviewers' comments:

Reviewer's Responses to Questions

**Comments to the Author**

1. Is the manuscript technically sound, and do the data support the conclusions?

Reviewer #1: Yes

Reviewer #2: Yes

2. Has the statistical analysis been performed appropriately and rigorously? 

Reviewer #1: Yes

Reviewer #2: I Don't Know

3. Have the authors made all data underlying the findings in their manuscript fully available?

Reviewer #1: Yes

Reviewer #2: Yes

4. Is the manuscript presented in an intelligible fashion and written in standard English?

Reviewer #1: Yes

Reviewer #2: Yes

5. Review Comments to the Author

Reviewer #1: The study highlights the potential of wastewater-based epidemiology (WBE) to go beyond merely quantifying variants by tracing the emergence and transmission pathways of SARS-CoV-2 variants. This is interesting work; here my suggestions.

The introduction lacks necessary citations to ensure that all previous work is cited properly.

Some references need to be updated to represent the literature accurately. For example, this sentence does not have the necessary citation diversity, and recent ones, “Several studies have successfully detected and identified SARS-CoV-2 variants in wastewater (Kuroiwa et al., 2023; Jahn et al., 2022; Layton et al., 2022).” Several recent studies identified the new variants through WBE, such as BA 2.86 and JN 1; therefore, these need to be cited because they lay the foundation for identifying SARS-CoV-2 new variants in WBE.

Also, lines 56 and 59, 68 to 71 need appropriate references added.

The study discussion is minimal and needs to be revised.

For example, EG.5 sub-variant C65601,XBB.1.16 sub-variant C65598, and BA.2.75 sub-variant C676 were detected only in Komatsu this can be due to WBE coverage differences in two cities 40% of Komatsu’s population and 57% of Hamamatsu’s ? these needs to be discussed.

The duration differences for variant prevalence between cities (e.g., 35 weeks in Komatsu versus 1 week in Hamamatsu) are interesting but require more in-depth discussion. This can be due to population diversity between cities or other factors that need to be discussed.

While the findings emphasize WBE's utility, the connection between wastewater data and real-world epidemiological trends is less discussed regarding variant landscape; for example, “some sub-variants that were not reported in clinical specimens were detected in wastewater," but there is no discussion of how these WBE findings are essential in terms of clinical utility or public health.

Acknowledge the study's limitations, such as sampling frequency, geographical scope, and reliance on wastewater data alone.

Reviewer #2: The authors have made a valuable contribution to the field of wastewater-based surveillance by highlighting the limitations of quantifying variant proportions, when it comes to tracing back the origin and transmission pathways of locally dominant variants. While wastewater-derived viral genome sequences represent a mix of multiple variants, they assume that each individual amplicon read is presumably originating from a single strain. Therefore, their insightful approach emphasizes the potential of haplotype analysis using partial viral genomes to more precisely trace the origin and transmission dynamics of SARS-CoV-2 variants. This work enhances our understanding of viral evolution in wastewater and provides a framework for more accurate epidemiological tracking.

However, the great potential of these findings is underserved by the lack of additional information. Here are questions that authors are invited to address and use to discuss the findings and provide additional clarifications to the study, as well as additional comments that are suggested to improve this work.

Minor points to be addressed

• Typos to be corrected:

Line 34: "which had never reported in clinical samples from Japan" should be changed to " which had never reported in clinical samples from Japan "

• Please remove highlights (references, supplementary data, figures) for the final version.

• Section "2.1. Wastewater sampling": the authors should mention the total number of samples collected.

• Lines 145-147: "To compare with known sequences sampled in Japan, we downloaded sequences of collection year 2023 from GISAID on 13 August, 2024". The authors should mention in the MM section the number of sequences downloaded.

• Lines 153-154: "Phylogenetic trees for the sub-variants were constructed using MEGA11 and the Neighbor-Joining method". No mention is made about the parameters used, if default options were chosen, etc. This applies to other tools mentioned and deserves revision.

• Figure 1: Panel C and Panel D have a number or missing data. Assuming that variants cannot suddenly disappear and stop circulating from one week to another in a sawtooth pattern, is this due to missing information, sample unavailability or no variant detected at all in those weeks?

Major points to be addressed

• Overall, the work done deserves better data interpretation and additional clarifications on the findings. Authors are invited to provide additional content to improve the manuscript readability and understanding (no reminder in the results section about the number samples sequenced, no discussion of data obtained as compared to results from other teams worldwide, etc).

• Authors have not mentioned the clinical cases reported at the same period of time the study was conducted within the two geographical areas analyzed, nor they have correlated their data with the clinical cases. This would help valorize their results and could give insights into how this approach could help detect variants before their clinical characterization.

• Authors should report the normalization method used if any, or correlate their data to known normalization options (please refer to Figure 1, Panel B).

• Authors should mention if factors such as environmental factors (viral RNA degradation, sample collection, or processing biases) could affect the reliability of haplotype analysis?

• Authors should discuss if integrating multiple approaches is feasible and/or could improve the accuracy of tracking viral variants in wastewater.

• How does haplotype analysis compare with other existing methods, such as whole-genome sequencing or computational deconvolution of mixed variants?

• The study assumes that each amplicon read originates from a single strain. However, mixed infections and recombination events could complicate this assumption. Authors should highlight how they assess the accuracy of this assumption. Does the method provide enough resolution to differentiate between closely related variants?

• Can this approach be applied universally, or are there constraints (sequencing depth, bioinformatics tools, computational requirements, etc) that may limit its implementation? As an example, how feasible is this method in low-resource settings?

• In terms of public health impact, how does this method improve real-time variant tracking for public health decision-making? Could it be integrated into routine wastewater surveillance programs for early warning of emerging variants?

Taking into account the points raised could help reinforce the quality of the approach and its impact in driving research in this area.

6. PLOS authors have the option to publish the peer review history of their article (what does this mean? ). If published, this will include your full peer review and any attached files.

**Do you want your identity to be public for this peer review?** For information about this choice, including consent withdrawal, please see our Privacy Policy .

Reviewer #1: No

Reviewer #2: No

---

## [Author Response · Author response to Decision Letter 1]

8 Apr 2025

All the responses are provided in the response letter.

---

## [Editor Report · Decision Letter 1]

10 Apr 2025

Dissemination of Local Sub-Variants of SARS-CoV-2 Detected by Detailed Mutation Analysis in Wastewater-Based Epidemiology

PONE-D-24-59114R1

Dear Dr. Honda,

We’re pleased to inform you that your manuscript has been judged scientifically suitable for publication and will be formally accepted for publication once it meets all outstanding technical requirements.

Kind regards,

Dhammika Leshan Wannigama, MD PhD

Academic Editor

PLOS ONE
---

## [Editor Report · Acceptance letter]

PONE-D-24-59114R1

PLOS ONE

Dear Dr. Honda,

I'm pleased to inform you that your manuscript has been deemed suitable for publication in PLOS ONE. Congratulations! Your manuscript is now being handed over to our production team.

Kind regards,

on behalf of

Dr. Dhammika Leshan Wannigama

Academic Editor

PLOS ONE